# Maternal folic acid supplementation and the risk of ankyloglossia (tongue-tie) in infants; a systematic review

Gal Rubin[1,2], Catherine Stewart[1], Laura McGowan[3], Jayne V. Woodside[3], Geraldine Barrett[1], Keith M. Godfrey[4], Jennifer Hall[1]*

1 UCL EGA Institute for Women's Health, Reproductive Health Research Dept, UCL, London, United Kingdom, 2 Minerva University: College of Natural Sciences, San Francisco, CA, United States America, 3 Centre for Public Health, Institute for Global Food Security, Queen's University Belfast, Belfast, Ireland, 4 MRC Lifecourse Epidemiology Centre and NIHR Southampton Biomedical Research Centre, University of Southampton and University Hospital Southampton NHS Foundation Trust, Southampton, United Kingdom

* jennifer.hall@ucl.ac.uk

## Abstract

### Background

Maternal folic acid supplementation is protective against the development of neural tube defects (NTDs) in babies. However, recent public-facing communications have raised concerns about a causal relationship between folic acid supplementation, particularly after the first trimester, and ankyloglossia (tongue-tie) in infants. Non-evidence-based communications are potentially harmful because they could adversely affect adherence to folic acid supplementation, increasing NTD occurrence. **This study aimed to** review evidence on the relationships between maternal folic acid supplementation during preconception and/or pregnancy and the risk of ankyloglossia in infants.

### Methods

We searched the databases MEDLINE, EMBASE, Cochrane CENTRAL, and Scopus. We searched for observational, and interventional studies, and systematic reviews investigating the effect of maternal folic acid supplementation during preconception or pregnancy on the occurrence of ankyloglossia in offspring. The search was registered on PROSPERO on 01/12/2022, ID: CRD42022375862.

### Results

The database searches yielded 93 articles. After removing duplicates and screening titles and abstracts, 26 remained. One article was judged relevant for inclusion in analyses; a case-control study that directly mentions the relationship between folic acid supplementation and ankyloglossia. **This study r**eported that regular intake of folic acid supplements was higher in women with infants with ankyloglossia. However, this study has limitations regarding design, selection bias, and confounding, calling the findings into question.

**Data Availability Statement:** All relevant data are within the paper and its Supporting Information files.

**Funding:** The author(s) received no specific funding for this work.

**Competing interests:** The authors have declared that no competing interests exist.

## Conclusions

Insufficient evidence exists for a relationship between folic acid supplementation and ankyloglossia. Currently, the benefits of folic acid supplementation far outweigh the risks. This must be clearly communicated to patients by their clinicians during preconception and antenatal care.

## Introduction

Maternal folic acid supplementation (FAS), when taken prior to and during pregnancy, is protective against the development of neural tube defects (NTDs) such as spina bifida and anencephaly, [1, 2] low birth weight, and megaloblastic anemia in mothers [3–6]. Annually, 260,100 NTD-affected pregnancies occur worldwide, constituting a significant part of the neonatal burden of care [3–6]. Fortification of staple foods with folic acid is common in numerous countries but where this has not been done, as in the United Kingdom (UK), FAS remains essential.

Ankyloglossia, or tongue-tie, is a condition that occurs when the tongue is tethered to the floor of the mouth by a short, thickened strip of tissue called the lingual frenulum [7, 8].This can make it difficult to move the tongue and may interfere with breast feeding. In some cases, ankyloglossia may also cause difficulty with speech [8]. Ankyloglossia can be treated with a minor surgical procedure, frenotomy, to release the lingual frenulum. The procedure is considered low-risk, in part as no general anesthetic is used [7, 9–12]. Ankyloglossia can also be treated non-surgically through speech therapy and lactation support [7].

The prevalence of ankyloglossia ranges from 2.8% - 10.7% globally [13, 14]; this variation is likely due, in part, to a lack of consensus on the diagnostic criteria. In recent years, there has been an increase in reported cases of ankyloglossia, primarily in high-income countries (HICs), thought to be due to an increased focus on breast feeding, a growing awareness of the condition, and the availability of treatment options [9]. However, in the non-academic literature, concerns have been raised among women of reproductive age and some health professionals in the UK and USA, of a causal relationship between FAS, particularly after the first trimester, and ankyloglossia (*Mothering.com*, *Netmums.com*). Currently, the clinical consensus is that FAS for preventing NTDs must begin at preconception [15]. Media reports suggesting a causal relationship between folic acid supplementation and ankyloglossia could affect FAS adherence, increasing NTDs.

This study aimed to conduct a systematic review of the scientific literature to identify and summarise evidence on the relationships between maternal FAS during preconception and pregnancy and the risk of ankyloglossia in the offspring.

## Materials and methods

### Search strategy and keywords

We searched the databases MEDLINE, EMBASE, Cochrane CENTRAL, and Scopus. The database searches were completed between January 19th and January 26th, 2023. Search terms covered synonyms for folic acid and different methods of administration or measurement all combined with the Boolean operator 'OR'. Similarly, synonyms for the periconception period and pregnancy were combined with OR, as were 'ankyloglossia' and 'tongue-tie'. A combination of MeSH terms and keywords was used depending on the database. These three searches were combined with 'AND'. In order to capture data related to folic acid supplementation that might be contained in studies about frenotomy (the surgical release of tongue-tie) that were

not picked up with the first search, we conducted a second search replacing the ankyloglossia terms with frenotomy. The full search terms are shown in S1 Appendix. The search was registered on PROSPERO on 01/12/2022, ID: CRD42022375862.

## Inclusion and exclusion criteria

We included observational, interventional studies, and systematic reviews assessing the relationships between folic acid and ankyloglossia in humans only. Studies had to present data on the effects of folic acid supplementation, either as a supplement or from the diet, separately from other (multi)vitamin supplementation. This could be assessed by the maternal recall of supplementation, assessment of diet, or analysis of blood folate levels. The outcome was the diagnosis of ankyloglossia in infants (aged up to 12 months). There were no language or date restrictions.

## Quality assessment

GR extracted data on study characteristics, design and results into a table designed for this review. Studies were assessed using the relevant Critical Appraisal Skills Programme checklist [16] by GR, CS and JH and discussed. No studies were excluded based on quality, but the findings of the critical appraisal were used to interpret the study's findings.

# Results

## Article selection

As shown in Fig 1 [17], the database searches yielded 73 articles after duplicates were removed. Titles and abstracts were critically reviewed by GR and JH against the inclusion criteria, and articles that were not relevant were excluded, leaving 26 articles for which full-text papers were retrieved and reviewed. One article remained for inclusion in analyses, as shown in Table 1. The reference lists of these articles were reviewed, but no additional studies were found.

There was only one study, conducted in Israel and published in 2020, directly assessing the relationship between FAS and ankyloglossia [18]. This case-control study reported that regular intake of FAS preconception was higher in women with infants diagnosed with ankyloglossia. Further details of the study are shown in Table 1. There were no studies looking at FAS after the first trimester.

## Quality assessment

Examination of the Amitai et al. study using the CASP (Critical Appraisal Skills Programme) case-control study checklist [16] showed a high risk of bias, calling the findings into question. This paper is limited by several factors, including participant selection, confounding variables, exposure and time frame measurements, and external validity.

Firstly, there is a lack of information regarding the recruitment strategy and selection methods for both the study and control groups; this lack of information makes it hard to determine the risk of selection bias within the study. There is also a lack of control for confounding factors. Specifically, there is no mention of matching cases and controls, with little information regarding the demographic characteristics of the two groups; as a result, we do not know if the control group was selected appropriately, or if the two groups were comparable. Furthermore, there is weak control for confounding factors within the analysis, with maternal age and folic acid intake being the only variables included in the logistic regression. As a result, it is uncertain whether the relationship found between folic acid intake and ankyloglossia is valid.

In addition, the measurements of exposure and time frames are unclear. Folic acid exposure was split into three levels: 'any', 'most', and 'regular'. While stratifying exposure into different

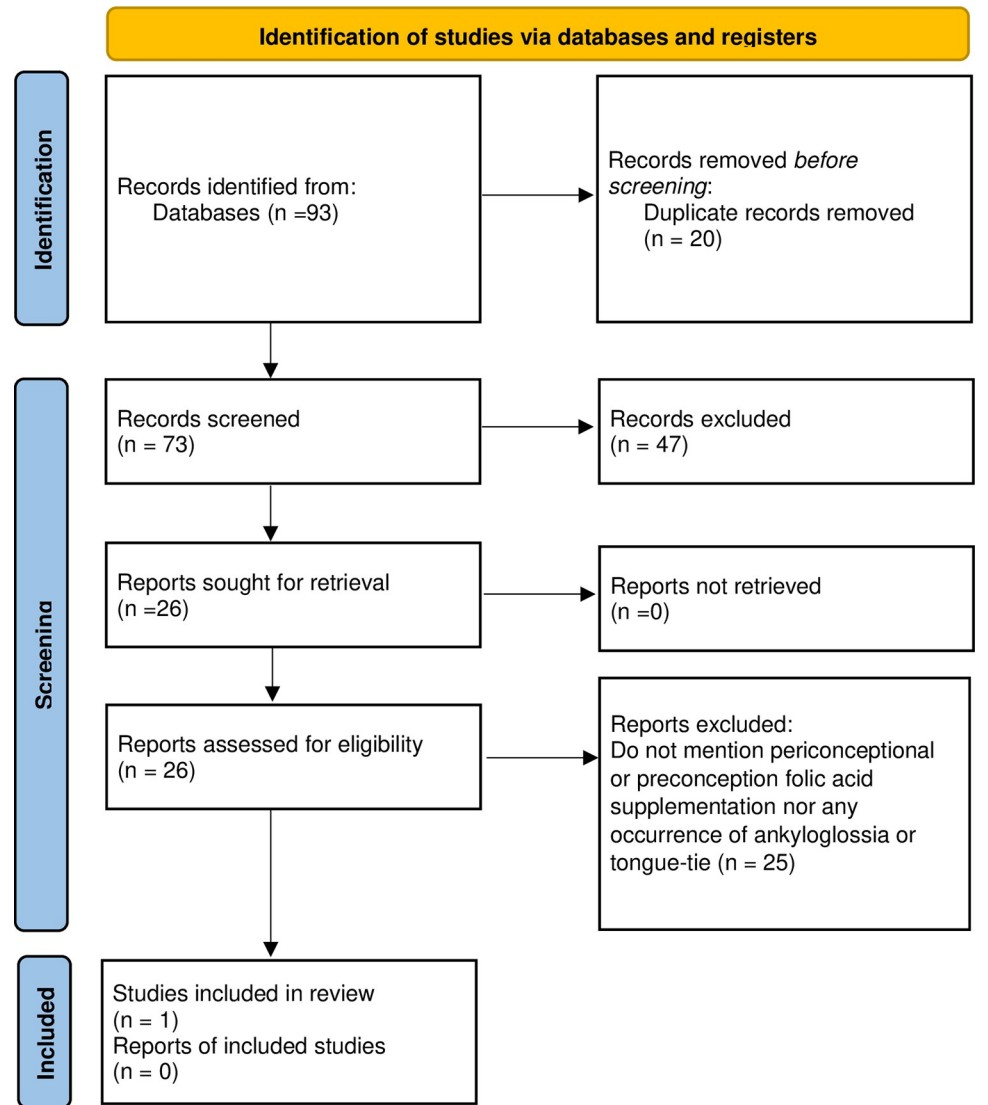

**Fig 1. Flow diagram: Article selection process.**

levels can often be useful, in this case, the authors do not clearly define the different levels of exposure (i.e., what is classed as 'most' vs 'regular' use) or how women understood this question; this is of particular importance as only the 'regular' use group was found to have a

**Table 1. Final article selection.**

| Author | Study | Sample size | Findings |
|---|---|---|---|
| Amitai et al., 2020. | Pre-conceptional folic acid supplementation: A possible cause for the increasing rates of ankyloglossia.<br>• Retrospective case-control.<br>• Israel. | 225 total.<br>• 85 infants with ankyloglossia.<br>• 148 infants in the original control group.<br>The study group was recruited from two pediatric clinics in Ramot, Jerusalem, and consisted of infants that underwent frenotomy for ankyloglossia. The control group was recruited from three different clinics in Jerusalem, and patients with ankyloglossia were excluded. | 'Regular' preconception folic acid intake was found to be higher in mothers of infants with ankyloglossia, compared to controls (OR 3.41, 95% CI 1.85–6.27, p-value < 0.0001). However, there was no significant difference found between the case and control groups for intake on 'most days' (OR 1.67, 95% CI 0.93–3.05, p-value = 0.07) or for 'any' intake (OR 1.45, 95% CI 0.77–2.78, p-value = 0.23). |

relation with ankyloglossia (OR 3.4, 95%CI 1.85–6.27, p<0.001). There is also a lack of information regarding the time frame of exposure, with no definition given for the pre-conception or peri-conception periods considered, meaning it is not clear how long women were taking folic acid for. Also, in terms of time frame, there is a lack of information regarding the time frame of ascertainment, with no information on how far back researchers were asking mothers to think, meaning the risk of recall bias could be high, especially since women were being asked to remember specific details of their folic acid intake.

Finally, due to the choice of the study group, and the choice of the study population, the results of this study are not generalizable. The study group was infants that had undergone frenotomy, suggesting a bias in ankyloglossia severity, while the study population was ultra-orthodox Jewish communities, meaning any results from the study would have uncertain external validity.

## Discussion

This systematic review of the literature shows that there is currently no clear evidence to support a relationship between folic acid supplementation after the first trimester and ankyloglossia. This is an important finding as it shows that there are no empirical grounds for the circulating media reports, to the contrary, such reports could adversely affect adherence to folic acid supplementation leading to an increase in NTDs. The only available evidence on a relationship between FAS and ankyloglossia was from a case-control study examining preconception folic acid intake, judged to have a high risk of bias through an inadequate selection process, incomplete reporting, and lack of clarity over the measurement of FAS intake and the definition of the 'periconception' period. Even if it had been well-conducted, as a case-control study it would rank lowly in the hierarchy of evidence, particularly given the lack of other supporting evidence.

There is, currently, no research evidence supporting a mechanism by which FAS might increase the risk of ankyloglossia in infants. Amitai et al. proposed that excessive folic acid during organogenesis (with no specific time frame specified) might result in tighter closure of mid-line structures, resulting in excess connective tissue at the base of the tongue, including the lingual frenulum [18]. One other potential mechanism has been proposed, relating to unmetabolized folates following folic acid supplementation [19]. However, in a randomized controlled trial testing the levels of plasma unmetabolized folates between women taking low and high doses of folic acid, no significant difference in unmetabolized folates was observed between the group [20].

Further research is needed to investigate the association between FAS (both before and during pregnancy) and ankyloglossia. Were any relationship found, subsequent research should explore the potential mechanisms by which folic acid supplementation could increase the risk of ankyloglossia in infants. If a link were identified this would then enable clinicians and women to make an informed decision around FAS, balancing the risk of the potentially life-threatening outcome of NTD and other known benefits of FAS on birthweight, child development, and cognition [21–23], with the risk of ankyloglossia in the infant if FAS is continued throughout pregnancy.

The timing of folic acid supplementation is essential for efficacy. Neural tube closure should occur by the fourth week of pregnancy; if the neural tube fails to close by then, malformations in the brain and spine, such as anencephaly and spina bifida, can occur [15]. Unfortunately, many people do not know they are pregnant until that window of opportunity has closed [2]. Therefore, to prevent NTDs, folic acid should be taken starting from at least one month before conception until the end of the first trimester to ensure that maternal folate levels are sufficient

in the early stages of pregnancy when neural tube closure occurs [24, 25]. It is difficult to get enough folate from diet alone [26]; around 80 countries, including the USA, Canada, Chile, and South Africa, have mandated the fortification of staple food products, such as corn and wheat, with folic acid to increase folate levels in the population and decrease adverse health outcomes [27, 28]. In 1998, in the USA, the FDA required that all enriched grain products be fortified with folic acid [29]. Since then, an estimated 1,300 NTDs have been prevented annually (about a 30% reduction in prevalence), saving an annual total of US$508 million [29]. Furthermore, studies analyzing the rates of congenital malformations after mandated wheat fortification suggest that it may be protective against congenital disabilities other than NTDs [3, 30–32]. In September 2021 the UK government decided to proceed with mandatory fortification of non-wholemeal wheat flour with folic acid. However, this decision has been met with criticism regarding the agreed-upon dosage, which is considered inadequate [27], and the policy has yet to be implemented.

In light of these large-scale results spanning several decades and countries, the evidence presented in the Amitai study does not suffice to change clinical practice and recommendations, and extensive, experimental, and clinical research is needed to establish a compelling causal link.

## Strengths and limitations

The strengths of this research are that we systematically searched a range of databases and used a variety of synonyms and related terms to capture all relevant data. We did not limit the search by time or language, to maximize the reach, and used broad inclusion criteria to allow us to include both observational and interventional studies. However, given the lack of a standardized definition and limited research on ankyloglossia, we were unable to find many studies directly relevant to our question. The only primary research we did find was of poor quality and had significant limitations.

## Conclusion

There is no strong evidence of a causal relationship between folic acid supplementation and ankyloglossia. At the current time, the known benefits of folic acid supplementation in the preconception period and during pregnancy far outweigh the potential risks, with overwhelming evidence to support supplementation before and during pregnancy. These benefits must be clearly communicated to patients by their clinicians during preconception and antenatal care.

## Supporting information

**S1 Checklist. PRISMA 2020 checklist.**
(DOCX)

**S1 Appendix.**
(DOCX)

## Author Contributions

**Conceptualization:** Laura McGowan, Geraldine Barrett, Keith M. Godfrey, Jennifer Hall.

**Formal analysis:** Gal Rubin.

**Investigation:** Gal Rubin.

**Methodology:** Catherine Stewart, Laura McGowan, Jayne V. Woodside, Geraldine Barrett, Keith M. Godfrey, Jennifer Hall.

**Supervision:** Jennifer Hall.

**Validation:** Jennifer Hall.

**Visualization:** Gal Rubin.

**Writing – original draft:** Gal Rubin, Catherine Stewart, Jennifer Hall.

**Writing – review & editing:** Gal Rubin, Catherine Stewart, Laura McGowan, Jayne V. Woodside, Geraldine Barrett, Keith M. Godfrey, Jennifer Hall.

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
