## [Decision Letter · Decision Letter 0]

25 Oct 2023

Maternal folic acid supplementation and the risk of ankyloglossia (tongue-tie) in infants; a systematic review

PONE-D-23-25110

Dear Dr. Hall,

We’re pleased to inform you that your manuscript has been judged scientifically suitable for publication and will be formally accepted for publication once it meets all outstanding technical requirements.

Kind regards,

Hanna Landenmark

Staff Editor

PLOS ONE

on behalf of 

Sagar Panthi, MBBS

Academic Editor

PLOS ONE

Additional Editor Comments (optional):

Reviewers' comments:

Reviewer's Responses to Questions

**Comments to the Author**

1. Is the manuscript technically sound, and do the data support the conclusions?

Reviewer #1: Yes

Reviewer #2: Yes

2. Has the statistical analysis been performed appropriately and rigorously? 

Reviewer #1: Yes

Reviewer #2: I Don't Know

3. Have the authors made all data underlying the findings in their manuscript fully available?

Reviewer #1: Yes

Reviewer #2: Yes

4. Is the manuscript presented in an intelligible fashion and written in standard English?

Reviewer #1: Yes

Reviewer #2: Yes

5. Review Comments to the Author

Reviewer #1: Thank you for the authors for their systemic review.

Despite the disappointing findings of having only one weak study handled the correlation between the Maternal folic acid supplementation and the risk of ankyloglossia, however this doesn't affect the hard work that you did throughout your project.

Reviewer #2: The different aspects of the manuscript are well written. The language is clear and unambiguous. The references are well written. However, in this review , only one published article was analyzed which may be a demerit. However, the manuscript is acceptable publication as it is.

6. PLOS authors have the option to publish the peer review history of their article (what does this mean?). If published, this will include your full peer review and any attached files.

Reviewer #1: **Yes: **Mena Abdalla

Reviewer #2: No

---

## [Editor Report · Acceptance letter]

27 Oct 2023

PONE-D-23-25110 

Maternal folic acid supplementation and the risk of ankyloglossia (tongue-tie) in infants; a systematic review 

Dear Dr. Hall:

I'm pleased to inform you that your manuscript has been deemed suitable for publication in PLOS ONE. Congratulations! Your manuscript is now with our production department. 

Kind regards, 

on behalf of

Dr. Sagar Panthi 

Academic Editor

PLOS ONE